# Novel Design of RNA Aptamers as Cancer Inhibitors and Diagnosis Targeting the Tyrosine Kinase Domain of the NT-3 Growth Factor Receptor Using a Computational Sequence-Based Approach

**DOI:** 10.3390/molecules27144518

**Published:** 2022-07-15

**Authors:** Ashraf M. Muhammad, Ali Zari, Nouf H. Alsubhi, Maryam H. Al-Zahrani, Rana Abdullah Alghamdi, Mai M. Labib

**Affiliations:** 1Applied Biotechnology Program, Faculty of Science, Ain Shams University, Cairo 11566, Egypt; 2Department of Biological Sciences, Faculty of Science, King Abdulaziz University, P.O. Box 80203, Jeddah 21589, Saudi Arabia; azari@kau.edu.sa; 3Biological Sciences Department, College of Science & Arts, King Abdulaziz University, Rabigh 21911, Saudi Arabia; nhalsubhi@kau.edu.sa; 4Biochemistry Department, Faculty of Science, King Abdulaziz University, Jeddah 21589, Saudi Arabia; mhsalzahrani@kau.edu.sa; 5Department of Chemistry, College of Sciences & Arts, King Abdulaziz University, Rabigh 21911, Saudi Arabia; raalghamdi3@kau.edu.sa; 6Department of Bioinformatics and Computer Networks, Agriculture Genetic Engineering Research Institute (AGERI), Cairo 12619, Egypt

**Keywords:** RNA modeling, aptamer design, molecular docking, in silico, SELEX

## Abstract

Aptamers, the nucleic acid analogs of antibodies, bind to their target molecules with remarkable specificity and sensitivity, making them promising diagnostic and therapeutic tools. The systematic evolution of ligands by exponential enrichment (SELEX) is time-consuming and expensive. However, regardless of those issues, it is the most used in vitro method for selecting aptamers. Therefore, recent studies have used computational approaches to reduce the time and cost associated with the synthesis and selection of aptamers. In an effort to present the potential of computational techniques in aptamer selection, a simple sequence-based method was used to design a 69-nucleotide long aptamer (mod_09) with a relatively stable structure (with a minimum free energy of −32.2 kcal/mol) and investigate its binding properties to the tyrosine kinase domain of the NT-3 growth factor receptor, for the first time, by employing computational modeling and docking tools.

## 1. Introduction

Aptamers, the single-stranded nucleic acid analogs of antibodies, hold a great promise in molecular diagnostics, therapeutics, and drug targeting, due to their sensitivity and high selectivity toward target molecules [1]. The small molecular weight of aptamers provides complex structures with sufficient surface areas to bind their targets with high specificity. Moreover, their ability to form unique and stable secondary structures allows them to create tertiary structures that recognize and bind their targets, and distinguish between the splice variants and post-translational modifications of the same protein [2].

Antibodies have long been applied in molecular diagnostics and clinical medicine [3,4]. However, antibodies have significant disadvantages in their production, such as their costly synthesis, batch-to-batch variation, cross-reactivity, and the possibility of contamination [1,5]. On the other hand, aptamers are easily synthesized and modified, with a high reproducibility [5,6]. Owing to their smaller size than antibodies, aptamers have high biocompatibility, low immunogenicity and toxicity, and better transport and tissue penetration properties. Generally, aptamers are more resistant to degradation than antibodies, and their stability can be further increased by chemically modifying the nucleotides to resist the endogenous nucleases [7]. Because they are chemically synthesized, the chemical modifications can be introduced at any desired position in the nucleotide chain [2]. The sugar modifications at the 2′ position (2′-amino, 2′-fluoro, and 2′-*O*-methyl) are one of the most common adjustments to improve the nuclease resistance [8]. To decrease the nonspecific binding, and make the synthesis process easier and more efficient, the aptamer length should be reduced as much as possible [7].

The systematic evolution of ligands by exponential enrichment (SELEX) approach is used to select aptamers in vitro, which involves the incubation of the target molecules with a pool of oligonucleotides, followed by a myriad of rounds of selection and enrichment to remove the unbound nucleotides and redeem the final aptamer pool [9,10,11]. As might be expected, this procedure is costly and time-consuming. Different in silico approaches and computational methods, such as modeling techniques and molecular dynamics (MD) simulations, are essential to reduce the time and cost of the selection and production of aptamers [10].

Bioinformatic tools are comprised of a broad range of methods that facilitate aptamer design and investigate their binding capabilities to the proposed targets. The molecular modeling involves the prediction of the secondary structure of RNA from its nucleotides sequence (primary system); then, building a 3D model (tertiary structure) based on this secondary structure [12]. The predicted 3D model can then be used in docking simulations to investigate different poses of the RNA-protein interaction and select the complexes with the lowest binding energies. Afterward, these in silico predictions can be validated through lab experiments to improve the binding properties of the aptamers and ensure the stability of the predicted structures. 

The NT-3 growth factor receptor (or tropomyosin receptor kinase C; TrkC) is an 839-amino acid transmembrane protein that features a 286-residue intracellular tyrosine kinase (TyrKc) domain [13]. Despite recent questions surrounding the TrkC-mediated oncogenesis mechanism, the overexpression of TrkC is known to drive cancer cell survival, migration, and metastasis [14]. Most notably, TrkC is overexpressed in neuroblastoma and glioma, and many other human tumors [14,15,16]. Given the advantages of aptamers for targeted therapy and the overexpression of TrkC in cancer, TrkC presents an exceptional target for aptamer design.

In this study, a high-affinity aptamer targeting the TyrKc domain of the NT-3 growth factor receptor was designed, for the first time, computationally, using a sequence-based approach. First, the TyrKc domain was docked with known aptamers present on the nucleic acid database (NDB) [17]. Then, the aptamer sequence with the highest affinity was used to generate new similar sequences by introducing random point mutations (substitutions) in the original sequence. Finally, the 3D models of these sequences were predicted and employed to assess their binding properties and validate their ability to inhibit the TyrKc domain through molecular docking simulations.

## 2. Materials and Methods

### 2.1. Virtual Screening to Identify Probable RNA Candidates

The aptamers were obtained from the NDB [17] by searching for RNA structures and selecting “RNA only” and “Aptamer” in the “Polymer” and “RNA Type” sections, respectively. The 3D structures were downloaded (Table 1) and prepared for docking by deleting undesired and repeated molecules using PyMOL software v 1.8 [18]. The PDB coordinates of the TrkC tyrosine-kinase domain (6KZC, resolution 2.00 Å, [19]) were taken from the Protein Data Bank and prepared by removing the water molecules, adding the polar hydrogens, and deleting the co-crystalized ligand. The dockings were performed globally, no binding site was specified, using the HDOCK web server to perform protein-protein and protein-DNA/RNA dockings, using a hybrid method combining template-based modeling and ab initio free docking [20,21,22].

### 2.2. Generation of New Aptamer Sequences

The sequence of the NDB aptamer with the highest affinity to the target domain was used to generate multiple sequences of the same length, by introducing random point mutations (substitutions) in the original sequence.

### 2.3. Molecular Modeling and Docking of the New Aptamers

The RNAfold and RNAComposer web servers were used to obtain the minimum free energy (MFE) tertiary RNA analogs of the generated sequences, to be used as ligands in the subsequent analyses [23,24,25,26]. The RNAfold generates the MFE secondary structure for each RNA sequence which is used as input by the RNAComposer to build the 3D model. Once again, the newly obtained structures were docked with the target domain to seek the best-bound aptamer.

### 2.4. Aptamer Optimization and Final Interaction Profiling

The best-bound aptamer was modified to get a more stable structure, and its interaction with the target domain was visualized in more detail to report the amino acids and nucleotides involved in the interaction, using the Protein-Ligand Interaction Profiler (PLIP) web tool [27].

## 3. Results

### 3.1. Virtual Screenings

Stronger interactions take place when more contacts are established between the ligands, aptamers in this case, and the target, to form a highly stable complex. From the obtained results, the 71-mer adenine riboswitch aptamer domain from *Vibrio vulnificus* (PDB: 5uza) had the highest binding affinity to the TyrKc domain (Table 2). Riboswitches are the regulatory segments of mRNA molecules (often <100 nt) that regulate the expression of the proteins, due to the binding of a ligand or a small molecule [28]. They comprise two fundamental domains: an aptamer domain that binds the ligand and an expression domain that controls the gene expression via various mechanisms [29].

### 3.2. Novel Aptamers Generation

The random mutations were introduced in the 5uza sequence, and ten new sequences were generated (Table 3); then, the secondary and tertiary structures were predicted (Figure 1). Some structures, such as seq 08 and 09, have relatively high MFE indicating that the structure might be unstable. 

### 3.3. Docking Results

The docking simulations assessed the interaction of the novel aptamers with the target domain. As shown in Table 4, four structures had a higher binding affinity to the domain than the main aptamer, indicating that the minor nucleotide variations could improve the aptamer structure stability and its binding properties, which lab experiments can further assess. Since seq 09, the best-binding structure, has a high free energy of −16.4 kcal/mol, the structure must be modified and optimized to obtain a more stable structure.

### 3.4. Optimization and Analysis of the Selected Aptamer

A smaller aptamer would have less interference from the steric effect and a lower synthesis field cost [4]. Consequently, some of the nucleotides of the seq_09 aptamer were modified or removed, yielding a 69-nt 3-hairpin modified structure (mod_09) of −32.2 kcal/mol. In addition, the overall structure has much less entropy than the original one (Figure 2). Notably, the modified structure formed a relatively less stable complex with the target domain (score: −591.9) than the original one (−673.4), but it is an acceptable result since it has a better score than the structure obtained from PDB (−590.1).

The interaction between the mod_09 aptamer and the target domain has three binding interfaces (Figure 3). The first binding interface is near the 5′ terminal of the aptamer employing two hydrogen bonds between the base G1 and the residue ARG745 and a salt bridge between G3 and ARG678. The second one has a larger surface area as it involves more interacting partners. A single hydrogen bond is formed between G53 and TYR669, in addition to the three salt bridges between bases U16, A17, and A18 with residues LYS818 and LYS822. The third binding interface has one salt bridge formed between G31 and LYS797, and two hydrogen bonds formed between the bases G29 and G30 with the residues GLN813 and GLN808, respectively (Figure 4).

## 4. Discussion

The aptamer selection with SELEX has multiple drawbacks that restrict its effectiveness and make it challenging to produce high-affinity aptamers. The SELEX method requires numerous rounds of selection and enrichment, which makes the process last for months [30]. Computational approaches are promising tools to overcome the time and costs of in vitro aptamer selection. Over the last 15 years, a wide range of computational methods have been utilized in designing aptamers [10,31]. The interest in in silico aptamer design is increasing, owing to the reduced usage of chemical reagents and wet-lab equipment and the relatively affordable costs of computational resources. The early studies on the computational design of aptamers aimed to reduce the initial number of oligonucleotides in the SELEX library pool and speed up the process of identifying the appropriate sequences. In this manner, a comprehensive approach was established to predict the structure of the RNA molecules and estimate their binding affinity to a target molecule through computational docking, however, this approach had multiple drawbacks, including the requirement of significant computational resources and the need for a considerably long time to generate the 3D structures for a single sequence [32]. Later, another technique that employed algorithms for motif scanning and screening was proposed by analyzing the RNA secondary structures, using a nucleotide transition probability matrix for pool generation and design [33]. In addition to the requirement for a rare parallel system supercomputer, this method was ineffective in discovering novel motifs and screening with tertiary structures, since the data were two-dimensional. A more time-saving workflow designed the aptamers from their 2D structures and modeled the 3D conformations by Rosetta, depending on the pre-docked mutated conformations and MD optimization, followed by a rigid docking procedure for the prediction of the RNA-protein complexes [34]. More recently, a computational approach, slightly similar to our study, was employed for the in silico design of aptamers, targeting the *Streptococcus agalactiae* surface protein [35]. The workflow comprised of the structural modification of a previously known RNA sequence, followed by the secondary structure prediction and the generation of 3D structures. The binding affinities were determined from the molecular docking simulations.

The present study aims to overcome the SELEX limitations by using a computational method to design aptamers targeting the TrkC tyrosine kinase domain. This study used a simple approach to select an aptamer binding the TyrKc domain of the NT-3 growth factor receptor. First, the known aptamers from the nucleic acid database were docked against the target domain to understand the possible binding sequences. Since this was an adenine-aptamer, it was quite unusual for it to have a high binding affinity to a protein domain. The adenine binding occurs within the aptamer structure between the bases themselves, while the binding of the proteins is more dependent on the outer interfaces. The best-bound sequence was modified to improve the binding properties, by introducing minor nucleotide variations. Consequently, ten new sequences were generated, four of which formed a more stable complex than that originated with the original structure from NDB. The top sequence (seq_09), unsuitably, had a high MFE value, indicating that the structure could be unstable. The structure was modified manually by replacing some of the bases and removing others to address this problem. As a result, a much more stable structure (mod_09) was obtained, as the MFE value decreased from −16.4 to −32.2 kcal/mol. Despite its relatively high stability, the mod_09 aptamer formed a lower stable complex with the target domain than its precursor. However, it can be considered an acceptable result since it was better than the original structure obtained from NDB.

Despite being simple, this method may contribute to the process of aptamer design in the future to reduce the number of sequences in the initial SELEX pool, by generating more sequences with similar nucleotides distribution or introducing minor mutations, or through coupling with other more sophisticated approaches.

## 5. Conclusions

In this study, the TyrKc domain of the NT-3 growth factor receptor was used, for the first time, as a target to demonstrate the potential of the bioinformatics methods and computational approaches as promising tools in the area of aptamer design and selection, by employing a complete set of in silico strategies for the development of aptamers, using a simple sequence-based procedure. As a result, a 69-nt aptamer was optimized to have a relatively stable structure and acceptable binding score to the target domain. Since this computer-based aptamer selection method is cost-effective, simple, and does not require sophisticated devices, it can be applied to obtain more stable aptamers or be implemented in other algorithms for aptamer design.

## Figures and Tables

**Figure 1 molecules-27-04518-f001:**
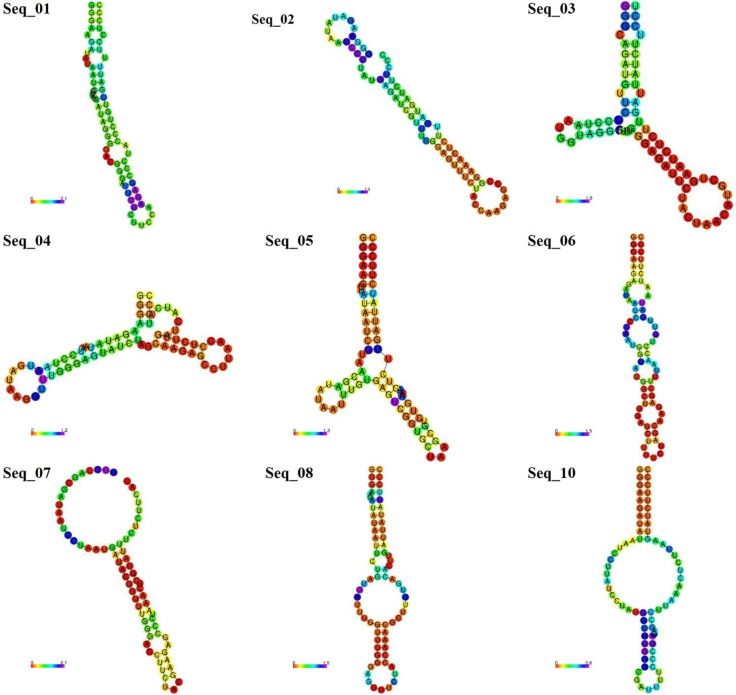
Secondary structures of the newly generated sequences from seq_1 to seq_10 (except seq_09). Nucleotides are colored according to their positional entropy, as shown on the horizontal bar. Red and orange colors indicate lower entropy, while blue colors indicate higher entropy.

**Figure 2 molecules-27-04518-f002:**
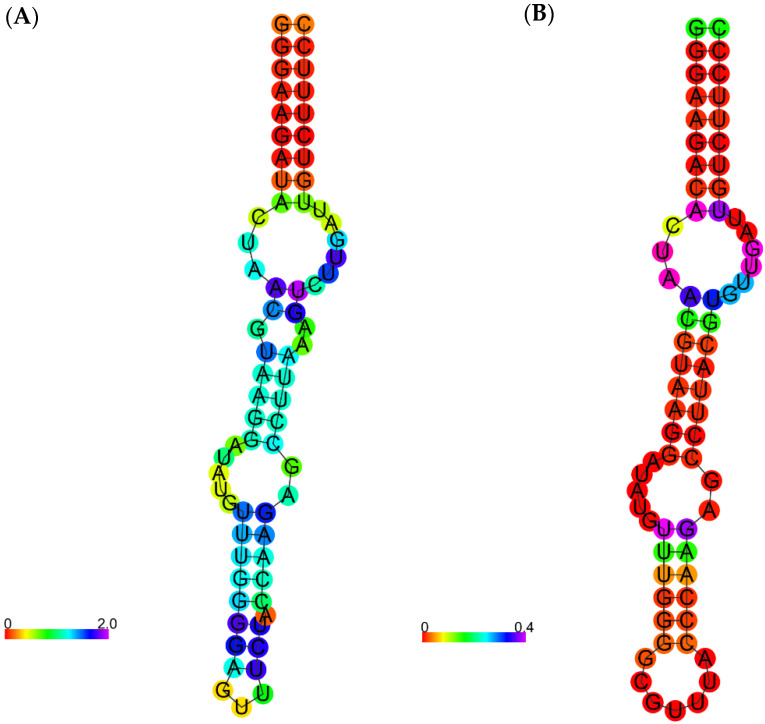
Secondary structures of (**A**) seq_09 aptamer and (**B**) the modified aptamer.

**Figure 3 molecules-27-04518-f003:**
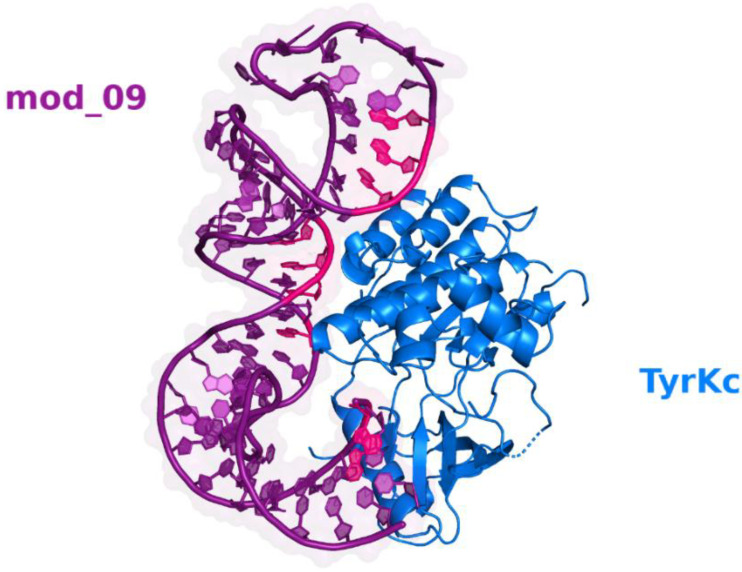
Cartoon view of the interaction showing the binding interfaces (pink) between the aptamer (purple) and the TyrKc domain (blue).

**Figure 4 molecules-27-04518-f004:**
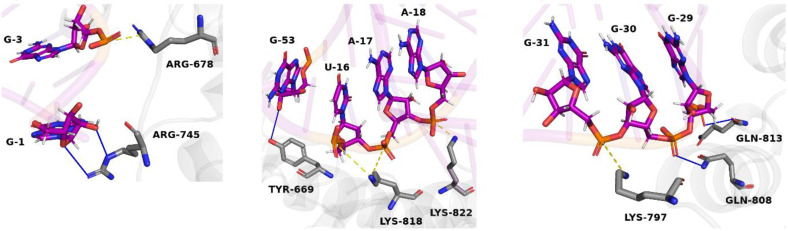
Cartoon view of the aptamer/protein complex, depicting the interacting nucleotides and amino acids in sticks view. Hydrogen bonds are shown in blue lines and salt bridges in yellow dashes.

**Table 1 molecules-27-04518-t001:** List of used PDB codes.

ID	ID	ID	ID	ID	ID	ID	ID	ID	ID
1ddy	1nta	2l1v	2rn1	4fe5	5bjo	6c63	6gzk	6ubu	7eog
1et4	1ntb	2lun	2rqj	4frg	5kpy	6c64	6k84	6up0	7kd1
1f1t	1q8n	2oom	2rrc	4l81	5lwj	6e80	6p2h	6v9d	7kvt
1nbk	2au4	2pn9	3gca	4lvv	5ob3	6e8s	6pq7	6vwt	7l0z
1nem	2jwv	2qbz	3q50	4oqu	5uza	6e8t	6q57	6wjr	7oa3

**Table 2 molecules-27-04518-t002:** Docking scores of the top ten binding aptamers to the target domain.

Aptamer	Docking Score
5uza	−590.1
6gzk	−576.3
5lwj	−532.8
2qbz	−512.5
2au4	−504.1
1q8n	−490.1
1nbk	−475.2
4lvv	−473.9
2lun	−465.9
2l1v	−463.4

**Table 3 molecules-27-04518-t003:** Newly generated sequences. Capital letters are the modified nucleotides.

Identifier	Sequence	Predicted MFE (kcal/mol)
5uza	gggaagauauaauccuaaugauaugguuugggaguuucuaccaagagccuuaaacucuugauuaucuuccc	−21.5
seq_01	gggaagauauaaucGuCaugauaGggGACgggaUuuucuUccaagagccCuaCCcuGuugauuUucCuccc	−21.1
seq_02	gggaagauauaauccuUaugaGauCguuugggaguuucuaccaagagccGGaaacucuugauGaucuuccc	−21.7
seq_03	gggCagauGuUCuccuaaugGuaGgguuugggagAuucuacUaaCaUGcuGaaUcucuugauuaucuuccU	−19.9
seq_04	gggaagauauaauccuaaugauaAgguuugggaguAucuaGcaagagccuuaaCcucuugauCaucAuccc	−17.8
seq_05	gggaagUuauaaucGuaaCgauauAAuuugUgaguCGGuGcUaagCgUcuGaaacucuugauuaucuuccc	−18.1
seq_06	gggaagaGauaauccCaaugGuauggCuGUAgCUuuucuaGcaagagcUuuaaCcuGuugauAaucuuccc	−21.5
seq_07	gggCagCGauaauccuaaugauaCgguuugggaCCuucuacGaagagccCuaaacGcGuAUuuCucuucAc	−17.2
seq_08	gggaagauauaauUcuGaugauUugguuugggaguuucuaccaaACgcUuuUGacAcuugauuauAuuccc	−15.2
seq_09	gggaagauaCUaAcGuaaGgauaugUuuGgggaguuucuaccaagagccuuaaaGucuugauuGucuuUcc	−16.4
seq_10	gggaagauauaauccuUauCCuaugguuugggCgAuuUuCccGagagccuuaaacucuuAaGuaucuuccc	−19.2

**Table 4 molecules-27-04518-t004:** Docking results of the newly generated structures.

Identifier	Docking Score
seq_09	−673.4
seq_10	−615.6
seq_06	−612.1
seq_02	−611.9
seq_08	−571
seq_04	−567.3
seq_01	−563.7
seq_03	−559.2
seq_05	−519.5
seq_07	−498.6

## Data Availability

The data that support the findings of this study are available from the corresponding author, A.M.M, upon request.

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
