# Peer review of "Novel Design of RNA Aptamers as Cancer Inhibitors and Diagnosis Targeting the Tyrosine Kinase Domain of the NT-3 Growth Factor Receptor Using a Computational Sequence-Based Approach"

_molecules, 2022, doi:10.3390/molecules27144518_

Round 1

Reviewer 1 Report

The paper presented by Gaafar et. al is very interesting for those dealing with aptameri. 

The ability to obtain aptamers with bioinformatics tools represents a great step forward and could have the ambition to maintain the entry of aptamers into clinical and diagnostic practice. 

I expect to read in the future that the experimental results confirm the goodness of the aptameri selected in this work.

Author Response

Appreciate your understanding and deep reviewing work an looking forwarded to apply it in more drugs discoveries   

Reviewer 2 Report

The article deals with in silico selection of nucleic acid aptamers. The topic is very acute and interesting for everyone working in the field of SELEX and applications of the aptamers, since proper computer simulations would save a lot of time and consumables to get the aptamer with desired affinity and specificity. The authors made a really good job with modeling and molecular docking. However, the overall study is highly disbalanced due to total absence of wet experiments. We cannot be sure about binding properties of the obtained aptamer until they are tested in real binding experiments. Therefore, I strongly recommend to prove the affinity of the resulting aptamer mod_09 by any of generally accepted methods (e.g,. SPR, BLI, MST or gel retardation assay). To my opinion, the article cannot be published without these experiments.

Besides, the choice of the parent aptamer, 71-mer adenine riboswitch aptamer domain from Vibrio vulnificus, also needs some discussion. It seems quite unusual that adenine-binding aptamer may possess a very different specificity to the TyrKc domain, a completely different aptatope. I think that the authors have to comment and discuss this issue.

Author Response

“However, the overall study is highly disbalanced due to total absence of wet experiments. We cannot be sure about binding properties of the obtained aptamer until they are tested in real binding experiments. Therefore, I strongly recommend to prove the affinity of the resulting aptamer mod_09 by any of generally accepted methods (e.g,. SPR, BLI, MST or gel retardation assay). To my opinion, the article cannot be published without these experiments.”

Defiantly the biomolecules interactions techniques such as Surface Plasmon Resonance (SPR), Isothermal Titration Calorimetry (ITC), Biolayer Interferometry (BLI), and Microscale Thermophoresis (MST) will add value to the work but our aim in this research to provide a computational model and Insilco analysis to predict the biomolecules interactions and open a window to complete these predictions using wet-lab experiments by other research groups, moreover, these strategies have been usefully applied with other groups such as Bruno et al 2021 in their work titled “Best Practices for Docking-Based Virtual Screening” https://doi.org/10.1016/B978-0-12-822312-3.00001-1, And zizeh Abdolmaleki et al 2021 “Use of Molecular Docking as a Decision-Making Tool in Drug Discovery” https://doi.org/10.1016/B978-0-12-822312-3.00010-2 that have been published in a very prestigious journal and were a key for a lot of discoveries.

“Besides, the choice of the parent aptamer, 71-mer adenine riboswitch aptamer domain from Vibrio vulnificus, also needs some discussion. It seems quite unusual that adenine-binding aptamer may possess a very different specificity to the TyrKc domain, a completely different aptatope. I think that the authors have to comment and discuss this issue.”

  • The riboswitch aptamer was chosen because it had the best binding score to the domain compared to the other aptamers, due to the high surface area involved in the interaction between the protein and the aptamer recruiting more bases than that involved in the binding with adenine. The original PDB structure (image) shows that adenine binds within the aptamer structure between the bases themselves, while the binding of proteins is more dependent on the outer interfaces.

Reviewer 3 Report

Is an interesting paper about of computational approaches to reduce the time and cost associated with the synthesis and selection of aptamers using several docking techniques. The paper is well written, and the experiments are well conducted. However, some points could be discussed.

In the present article the authors put forward a methodology based on computational techniques to design an aptamer able to bind the tyrosine kinase domain of the NT-3 growth factor receptor. Although the methodology sequence is logical, the process is not strong enough to be considered an alternative to aptamer design. The use of more accurate techniques would be necessary to achieve the goal.

In the introduction section, the authors highlight the importance of molecular dynamics during the selection process of aptamers, however in their methodology they did not include the tLEaP MDMix methodology to search cosolvent box to identify solvent sites and determine probable ligand binding sites. In addition, for the docking experiments, the author only used HDOCK web server. The authors could explain why they not considered use Ribodock instead? as this software is specific for RNA molecules. The authors claim that this methodology is novelty, but several approaches have been used such as AptaMut, Rtools and COMPAS for the analysis, design, identification, or clustering of the aptamers, none of this methodology was stated in this work. The experiments need to be validated in an in vitro model.

Figure 4 must be improved. Instead of the surface view, represent the image as cartoon-stick or cartoon-line to identify interactions.

Minor revision:

Check spelling and grammar, some words are wrongly used along the manuscript.

Line 18. The word “intriguing” is misused in this sentence. Consider replacing it.

Line 23: Delete the “s” from the word aptamers.

Line 50-51. Check grammar of the sentence. 

Line 55. Figure 1 does not provide additional information to the work. Indeed sentence from lines 52-55 is clear enough to be understood without the need of the figure.

Line 122-124. The authors state that the 5uza aptamer showed the highest binding affinity, but the data is not provided into the manuscript. Include a supplementary table with such information.

Lines 98-102. For the docking methodology, the authors did not use tLEaP MDMix methodology to look for a cosolvent box, thus, the results obtained may not be relevant when the ligand is absent.

Finally, I consider the manuscript could be improved and major changes are needed.

 Is an interesting paper about of computational approaches to reduce the time and cost associated with the synthesis and selection of aptamers using several docking techniques. The paper is well written, and the experiments are well conducted. However, some points could be discussed.

In the present article the authors put forward a methodology based on computational techniques to design an aptamer able to bind the tyrosine kinase domain of the NT-3 growth factor receptor. Although the methodology sequence is logical, the process is not strong enough to be considered an alternative to aptamer design. The use of more accurate techniques would be necessary to achieve the goal.

In the introduction section, the authors highlight the importance of molecular dynamics during the selection process of aptamers, however in their methodology they did not include the tLEaP MDMix methodology to search cosolvent box to identify solvent sites and determine probable ligand binding sites. In addition, for the docking experiments, the author only used HDOCK web server. The authors could explain why they not considered use Ribodock instead? as this software is specific for RNA molecules. The authors claim that this methodology is novelty, but several approaches have been used such as AptaMut, Rtools and COMPAS for the analysis, design, identification, or clustering of the aptamers, none of this methodology was stated in this work. The experiments need to be validated in an in vitro model.

Figure 4 must be improved. Instead of the surface view, represent the image as cartoon-stick or cartoon-line to identify interactions.

Minor revision:

Check spelling and grammar, some words are wrongly used along the manuscript.

Line 18. The word “intriguing” is misused in this sentence. Consider replacing it.

Line 23: Delete the “s” from the word aptamers.

Line 50-51. Check grammar of the sentence. 

Line 55. Figure 1 does not provide additional information to the work. Indeed sentence from lines 52-55 is clear enough to be understood without the need of the figure.

Line 122-124. The authors state that the 5uza aptamer showed the highest binding affinity, but the data is not provided into the manuscript. Include a supplementary table with such information.

Lines 98-102. For the docking methodology, the authors did not use tLEaP MDMix methodology to look for a cosolvent box, thus, the results obtained may not be relevant when the ligand is absent.

Finally, I consider the manuscript could be improved and major changes are needed.

Author Response

“Although the methodology sequence is logical, the process is not strong enough to be considered an alternative to aptamer design. The use of more accurate techniques would be necessary to achieve the goal.”

  • Indeed, there are more accurate techniques based on more sophisticated approaches, but all these methods are still proposed. There is no straightforward or standard method for designing aptamers computationally. All methods, including mine, would need further investigations to determine which one is more reliable to use. So, I aimed to introduce a new method that may contribute to the process in the future either directly or through coupling with other methods.

“In the introduction section, the authors highlight the importance of molecular dynamics during the selection process of aptamers, however in their methodology they did not include the tLEaP MDMix methodology to search cosolvent box to identify solvent sites and determine probable ligand binding sites.”

  • In the introduction paragraph 4 “Bioinformatics tools are comprised of… “, I intended to give an overall view of the tools that may or may not be included in the computational design of aptamers just like giving a note on the experimental SELEX process in the previous paragraph. The employed methodology in our study is stated clearly in the last paragraph of the introduction. Despite, the sentence was removed from the manuscript to avoid confusion.

“In addition, for the docking experiments, the author only used HDOCK web server. The authors could explain why they not considered use Ribodock instead? as this software is specific for RNA molecules.”

  • As stated on their website (link here), rDock (previously RiboDock) is used for docking small molecules against proteins and nucleic acids, not for docking two macromolecules. While HDOCK (link here) is optimized for Protein-protein and protein-DNA/RNA docking.

“The authors claim that this methodology is novelty, but several approaches have been used such as AptaMut, Rtools and COMPAS for the analysis, design, identification, or clustering of the aptamers, none of this methodology was stated in this work.”

  • These programs are used in the analysis of SELEX experimental data for high-throughput sequencing, which is stated in the following articles.

https://www.ncbi.nlm.nih.gov/pmc/articles/PMC4345306/

https://www.ncbi.nlm.nih.gov/pmc/articles/PMC5751119/

“The experiments need to be validated in an in vitro model.”

our aim in this research is to provide a computational model and Insilco analysis to predict the biomolecules interactions and open a window to complete these predictions using wet-lab experiments by other research groups, moreover, these strategies have been usefully applied with other groups such as Bruno et al 2021 in their work titled “Best Practices for Docking-Based Virtual Screening” https://doi.org/10.1016/B978-0-12-822312-3.00001-1, And zizeh Abdolmaleki et al 2021 “Use of Molecular Docking as a Decision-Making Tool in Drug Discovery” https://doi.org/10.1016/B978-0-12-822312-3.00010-2 that have been published in a very prestigious journal and were a key for a lot of discoveries.

“Lines 98-102. For the docking methodology, the authors did not use tLEaP MDMix methodology to look for a cosolvent box, thus, the results obtained may not be relevant when the ligand is absent.”

  • The addition of solvent box is relevant to Molecular dynamics simulations which were not used in the study.

https://pubs.acs.org/doi/10.1021/acs.jcim.1c00134

  • All grammar and word checking was done, so as the editing on Figure 4, the removal of Figure 1 and the addition of the table with the binding affinities (Table 2).

Round 2

Reviewer 2 Report

In silico works are indeed very important and gives a lot of possibilities for design of novel aptamers. Nevertheless, an overwhelming majority of studies on in silico selection and design of aptamers include experimental proving of their binding properties as a mandatory part. See, for example, a review of Buglak et al. (doi:10.3390/ijms21228420). I am still sure that the obtained aptamer must be characterized experimentally. Without these experiments, we cannot consider the presented computational as valid, amd this makes the whole article inconclusive. 

Author Response

“In silico works are indeed very important and gives a lot of possibilities for design of novel aptamers. Nevertheless, an overwhelming majority of studies on in silico selection and design of aptamers include experimental proving of their binding properties as a mandatory part. See, for example, a review of Buglak et al. (doi:10.3390/ijms21228420). I am still sure that the obtained aptamer must be characterized experimentally. Without these experiments, we cannot consider the presented computational as valid, and this makes the whole article inconclusive.”

  • Wet-lab experiments are indispensable for validating the computational-derived methods. Although, it is not essential to involve lab experiments in every computer analysis. One of the main advantages of bioinformatics is the fast growth and improvement achieved and how it can generate a large amount of data in less time than lab experiments.

As a matter of fact, in the review of Buglak et al, there are multiple studies performed fully based on computational methods including..

Reference 39 (https://doi.org/10.1016/j.matpr.2019.06.097)

Reference 48 (https://doi.org/10.1016/j.jsb.2015.07.003)

Another fully computational method was introduced in this article

(https://doi.org/10.1016/j.compbiolchem.2015.06.005)

Reviewer 3 Report

The authors make the corrections and they improve the paper, the discussion about of the methodology points is fine, however they could discussed more about of this points in the paper, just to offer an overview between different methods and why is more convenient one method over other. 

Author Response

“The authors make the corrections and they improve the paper, the discussion about of the methodology points is fine, however they could discussed more about of this points in the paper, just to offer an overview between different methods and why is more convenient one method over other.”

  • DONE, other methods are summarized in the discussion (1st paragraph)